# Mountain Hiking: Prolonged Eccentric Muscle Contraction during Simulated Downhill Walking Perturbs Sensorimotor Control Loops Needed for Safe Dynamic Foot–Ground Interactions

**DOI:** 10.3390/ijerph20075424

**Published:** 2023-04-06

**Authors:** Inge Werner, Francisco J. Valero-Cuevas, Peter Federolf

**Affiliations:** 1Department of Sport Science, Universität Innsbruck, 6020 Innsbruck, Austria; 2Division of Biokinesiology & Physical Therapy, University of Southern California, Los Angeles, CA 90089, USA; 3Alfred E. Mann Department of Biomedical Engineering, University of Southern California, Los Angeles, CA 90089, USA

**Keywords:** leg dexterity, medio–lateral balance, ante–posterior balance, declined treadmill walking

## Abstract

Safe mountain hiking requires precise control of dynamic foot–ground interactions. In addition to vision and vestibular afferents, limb proprioception, sensorimotor control loops, and reflex responses are used to adapt to the specific nature of the ground contact. Diminished leg dexterity and balance during downhill walking is usually attributed to fatigue. We investigated the supplementary hypothesis that the eccentric contractions inherent to downhill walking can also disrupt muscle proprioception, as well as the sensorimotor control loops and reflex responses that depend on it. In this study, we measured leg dexterity (LD), anterior–posterior (AP) and medio–lateral (ML) bipedal balance, and maximal voluntary leg extension strength in young and healthy participants before and after 30 min of simulated downhill walking at a natural pace on a treadmill at a 20° decline. Post–pre comparisons of LD (*p* < 0.001) and AP balance (*p* = 0.001) revealed significant reductions in dynamic foot–ground interactions after eccentric exercise without an accompanying reduction in leg extension strength. We conclude that eccentric contractions during downhill walking can disrupt the control of dynamic foot–ground interactions independently of fatigue. We speculate that mountaineering safety could be improved by increasing conscious attention to compensate for unadjusted proprioception weighting, especially in the descent.

## 1. Introduction

Mountain hiking is a popular recreational sport benefitting cardiovascular and muscular fitness. Injury risk and accidents counteract its health benefits, with falls in particular representing the second most frequent reason for mountaineering injuries [1,2]. Up to 75% of these falls happen during descent [2,3]. The impacts and trauma of falls in steep terrain may explain this injury prevalence. While fatigue during descent can reasonably be considered an important contributing factor, Faulhaber et al. [2] reported that people who suffered falls in the mountains rated their fatigue at the time of the accidents at an average of 2.4 ± 2.3 out of 10. To improve hiking safety, therefore, it is necessary to identify other potential contributors to falls, such as stepping inaccuracy or unexpected slipping. We explore the role of deficits in the sensorimotor control of dynamic foot–ground interactions, as it is a mechanism for stepping inaccuracies, trips, and slips.

Strength, multi-joint coordination, and sensorimotor processing are independent contributors to overall balance ability [4]. Multi-joint coordination and sensorimotor processing rely on visual, vestibular, tactile, and proprioceptive afferents. These multiple afferent channels are weighted and combined at different time scales to inform the sensorimotor control loops and reflex responses that produce motor commands to achieve dynamic balance [4,5,6]. Research on postural stability in a perturbed stance identified short-, medium-, and long-latency responses modulated by different parts of the central nervous system [7,8,9]. Recent studies have shown a process of reweighting sensory input to maintain motor performance when muscle proprioception is disturbed by tendon vibration [6,10,11] or external vestibular stimulation [12]. This raises the question whether eccentric muscle activation—which occurs during downhill walking [13,14] and is known to disrupt muscle spindle and Golgi tendon organ proprioception [15,16]—could cause the reweighting of sensory inputs and, therefore, influence movement control [16,17].

Movement control consists of two strategies: feed-forward (anticipative) and feedback (reactive) mechanisms. Specifically, the onset of stepping triggers feed-forward mechanisms, such as muscle pre-activation or co-contraction, as well as enhanced stretch reflex activity. Both mechanisms are employed to adapt joint stiffness in unstable situations at foot–ground contact [18]. Upregulation of co-contraction of leg muscles is expected when unstable foot–ground interactions must be controlled [19]. The skill of controlling unstable foot–ground interactions can be defined as leg dexterity and specific measurement devices have been developed to quantify this ability [4,20,21].

Furthermore, the activation of rapid reflex loops triggered by proprioceptive signals can be observed in balance tasks on unstable surfaces [7,22,23]. While the effects of fatiguing eccentric exercise on changes in proprioception and joint position sense are well-studied [24,25], research on non-fatiguing protocols is lacking. Prolonged eccentric muscle activity, as experienced during downhill walking, might interfere with these proprioceptive mechanisms and, therefore, contribute to heightened fall risk in descents [2,17,26].

The aim of the current study was to investigate the hypothesis that the continuous, cyclic eccentric loading inherent to light-effort downhill walking disrupts muscle proprioception afferents. Distribution to muscle afferents, in turn, perturbs sensorimotor control loops and reflex responses, causing a decrease in the performance of the functions that depend on it. This hypothesis was tested in two experiments, with the first quantifying the effect of simulated downhill walking on leg dexterity and the second, conducted with a separate cohort of subjects, quantifying the impact on double-limb balance and leg extension strength. Subjective fatigue was monitored in both experiments through the rate of perceived exertion (RPE).

## 2. Materials and Methods

### 2.1. Participants

In a series of pilot studies, we determined that medium-to-high effect sizes could be expected for the main dependent variables. Power analyses (g*power 3.1 [27], α = 0.05) suggested that sample sizes of 20 individuals in each group would yield a power greater than 0.80. The first experiment included 19 young and physically active participants with a mean age of 22.8 ± 1.9 years (6 female, 13 male, 1 dropout). The second study included a separate cohort of 40 young and physically active adults with a mean age of 25.3 ± 2.9 years (20 female, 20 male). Half of the subjects (gender-balanced) served as controls. 

The experiments were conducted in accordance with the Declaration of Helsinki. Test protocols applied were approved by the local Review Board of the University of Innsbruck (Nr. 0552012, Nr. 0632013). All participants were free of leg injuries and pain. Before participating in the study, participants signed a written informed consent after the study was described to them.

### 2.2. Experimental Design

Simulated downhill walking was performed on a treadmill set at a decline of 20° (Pulsar, h/p/cosmos, Germany). Walking speed was controlled in relation to each participant’s leg length [28] to provide a comfortable walking speed (1.47 to 1.61 m/s). A bout of 30 min at these comfortable walking speeds was chosen to assure low exercise intensity [29] while providing prolonged eccentric exercise, particularly to the quadriceps group. We measured heart rate (HR) every five minutes and at the end of treadmill walking. In addition, we asked subjects to report their level of exertion (at the beginning and immediately after the exercise bout) using the BORG scale (RPE), which ranges from 6 to 20 [29].

#### 2.2.1. Experiment One

“Leg dexterity” (LD) is the ability to stabilize unstable, dynamic foot–ground interactions [4,20,21]. In our study, it was measured with the Valero Dexterity Test^®^ (Neuromuscular Dynamics, LLC, La Crescenta, CA, USA) [4,20,21]. The LD test consists of asking seated subjects to use one foot to compress a platform atop a calibrated slender spring prone to buckling. Participants stabilize their body posture and the other leg by sitting on a bicycle saddle, placing the other foot firmly on the ground, and supporting their upper body by grabbing onto a fixed metal bar. The hip and knee of the active leg are flexed to approximately 75°. The more the foot presses on the platform, the more the spring tends to buckle and the more unstable the compression becomes. The spring is designed to buckle at low forces (~15% of body weight). This emerging instability needs to be stabilized by dynamic sensorimotor control at low leg forces using short- and medium-latency sensorimotor control loops. Subjects are instructed to press down and hold the maximal level of instability they can control for about 10 s. The downward force on the spring is, therefore, a quantitative measure of their maximal ability to stabilize an unstable foot–ground interaction [4,20,21]. The higher the force that can be applied and sustained, the better the leg dexterity is. Subjects performed three series containing three 10 s trials of compressions using their right leg. The maximum of the mean vertical force in Newtons applied to the spring quantifies the amount of instability the subject can control. The leg dexterity test is highly reliable (ICC = 0.94) [20] and uses force values far below leg extension strength maxima.

In the first session, participants familiarized themselves with the leg dexterity test. In the following testing session, participants performed the leg dexterity test after an individualized warm-up followed by 30 min of downhill walking where HR and RPE were monitored. After treadmill walking was terminated, participants walked a few steps on level ground to the leg dexterity device and performed the test again.

#### 2.2.2. Experiment Two

Balance ability was tested using the MFT-S3 check test [30,31]. The test requires quiet bipedal standing without shoes on a platform that can tilt in the medio–lateral or the anterior–posterior directions. The arms were allowed to move freely. After a 15 s familiarization, two measurement time blocks, each lasting 20 s, were recorded with a 30 s rest period in between. Each measurement time block resulted in a stability score representing duration in the tilted configuration, as well as tilt changes (for details, see [31]). The better score from each of the two trials was selected for each participant. The reliability of the MFT-S3 check is rated as high (ICC > 0.93) [31] for both the medio–lateral (MFT_ML) and the anterior–posterior (MFT_AP) tests. The lower the score is, the better the balance performance. The intervention group (INT) was familiarized with balance tests in a first session and performed the isometric maximum strength test for knee extensors in the dominant leg at a knee angle of 90°. In a separate intervention session, a short warm-up preceded the MFT_ML and MFT_AP tests. After 30 min of downhill walking, as described in the experimental design above, participants repeated the MFT_ML and MFT_AP tests and we measured their maximal knee extensor strength with the dominant leg. The control group (CON) conducted balance tests with 30 min of rest between the two test applications.

### 2.3. Statistics

The average of the best three maximal sustained force scores with the leg dexterity device and the maximal leg extension strength pre- and post-downhill walking were compared using a paired, two-tailed Student’s t-test. We confirmed our assumption of normality by applying the Shapiro–Wilk test. Cohen’s d effect size was calculated, with effects of 0.2–0.5 being considered small, 0.5–0.8 as moderate, and greater than 0.8 as large [32]. To compare balance scores in experiment two, a repeated measures ANOVA (pre–post) with the grouping factor CON-INT was conducted for MFT_ML and MFT_AP separately. Effect sizes are reported utilizing partial eta squared, with effects below 0.01 being considered small, 0.06–0.14 as moderate, and greater than 0.14 as large [32]. All values are expressed as means ± standard deviation. Statistical significance was set to α = 0.05, and we applied Bonferroni–Holm corrections to post hoc tests. All statistical calculations were undertaken in SPSS (IBM, Armonk, NY, USA, version 25).

## 3. Results

Leg dexterity (maximal force score) decreased significantly after downhill walking with a large effect size (t(18) = 5.5, *p* < 0.001, d = 1.3) (see Figure 1a). HR slightly increased from 86 ± 10 to 99 ± 12 bpm (*p* < 0.001) and RPE increased from 9.0 ± 2.3 to 12.5 ± 1.7 (out of 20, representing a light effort on the scale; *p* < 0.001).

Balance scores in the anterior–posterior direction (MFT_AP test) showed a significant interaction of repetition by group with a high effect size (F(1,38) = 13.3, *p* = 0.001, η^2^_p_ = 0.26). Post hoc tests showed a significant difference in balance scores for MFT_AP after exercise (INT) compared to CON (post-exercise, INT vs. CON: t(38) = 4.1, *p* < 0.001, d = 1.3) and a significant decrease in balance ability from pre to post for INT (t(19) = 2.8, *p* = 0.012, d= 0.6) with a medium effect (see Figure 1b). In the medio–lateral direction (MFT_ML test), no effects from the downhill-walking intervention were found (interaction of repetition by group *p* = 0.133). Knee extensor isometric strength did not change post-exercise (*p* = 0.511). Again, HR increased from 89 ± 6 to 95 ± 8 bpm and RPE increased from 8.4 ± 1.8 to 11.7 ± 2.6 (out of 20, representing a light effort on the scale; *p* < 0.001 respectively). 

## 4. Discussion

Our results show (i) a deterioration in leg dexterity and (ii) a decrease in balance performance in the anterior–posterior direction but not in the medio–lateral direction after 30 min of simulated downhill walking. HR and RPE increased slightly but did not exceed the “light effort” level of the RPE scale. Peripheral fatigue in the leg muscles was not detectable in terms of maximal voluntary knee extension force. As a result, dexterity scores were not biased by maximum muscle activation. Thus, downhill walking in our study could be considered a low-load exercise. At the same time, the predicted effects on dexterity and balance performance were confirmed with large effect sizes. Given that downhill walking is known to affect excitability of muscle proprioception afferents, our results show a decrease in the performance of the functions (leg dexterity and postural control) that depend on proprioception, which could be explained by perturbed sensorimotor control loops and reflex responses. Several studies have used interventions leading to muscular fatigue and documented combined deficits in proprioception [24,33] and postural control [34,35]. However, we only found one other study of balance impairment after a light effort exercise [36]. The study found lower performance in a one-leg stance on an unstable platform after downhill walking [36]. Our results corroborate and critically extend those findings.

Muscle activity generated against gravity, particularly extension torque at the knee, increases when walking downhill, whereas hip and ankle extensor activity do not increase, indicating different muscle coordination patterns in uphill, downhill, and level walking [37]. Body position at heel strike differs when comparing uphill and downhill walking. Walking downhill provokes an open hip angle, whereas knee angle and ankle dorsiflexion at initial ground contact stay unchanged [13]. Ground contact initializes a weight absorption pattern, especially at the ankle and hip, to lower the center of mass for controlled downward stepping [13,14]. Differences in muscle recruitment structure when walking downhill suggest a different sensory weighting for controlling downhill stepping, including muscle spindle afferents of eccentrically contracting muscles. Indeed, 15 min of isolated mild eccentric ankle activity did not compromise leg dexterity. Neither did 15 min of level walking nor 30 min of rest [17]. In contrast, the intervention in the present study required the whole gait movement pattern to adapt to downhill walking, especially with the eccentric use of knee extensors [37], and caused reductions in leg dexterity scores.

No effect from downhill walking was observed when medio–lateral tilting of the support surface had to be controlled (MFT_ML). It seems that inter-limb coordination was not compromised and repositioning of the trunk in the medio–lateral direction used unaffected sensorimotor control loops to meet the balance task. Although leg extensors, as well as hip muscles, are involved in downhill walking, the relevant proprioceptive signals to succeed in the MFT_ML task seem to be unchanged in signal sensitivity and weighting [38]. In contrast, results for the MFT_AP task revealed a significant decrement in postural control and balance after 30 min of downhill walking. Walking, especially downhill, is a dynamic task requiring anterior–posterior position control while monitoring medio–lateral body shifts. Differences in muscle afferents, and, therefore, differences in feedback integration, between treadmill walking and stance control in the anterior–posterior direction can be assumed to explain the study results [38,39]. Eccentric exercise fosters downregulation of the discharge rate in lengthening muscles (e.g., soleus, gastrocnemius, or quadriceps muscles) and modulates signaling pathways, mainly at the spinal level [16,39]. Spinal level contributions, such as short-latency responses, build a basis for balance control, especially in high velocity displacements of the foot [40], and produce adaptations after balance training [41]. Similarly to the close relationship found in this study between anterior–posterior balance control and leg dexterity, a recent study showed that training leg dexterity (with the same leg dexterity device) significantly reduced anterior–posterior center-of-pressure excursions during a single-limb stance on a rigid surface [42].

Although participants took steps on stable ground after treadmill walking, the activation of reweighting sensory information seemed to be delayed long enough to affect our outcome variables. In experiments using tendon vibration, recovery was observed within a time frame of 10 to 20 s [43] compared to more than 5 min in our experimental design. Prolonged, monotonous—but not fatiguing—eccentric muscle activation seems to provoke sustained maladaptation. Being aware of this time delay, more conscious postural control in descents might favor long-latency responses and could offer a mitigation strategy.

Even though both experiments resulted in significant changes, performance differences were small. The mean leg dexterity force decreased less than 10 N (although the nonlinear nature of the task prevented a simple linear comparison). Similarly to this finding, MFT_AP mean scores before and after downhill walking remained in the same norm category [31]. In summary, our results suggest that control of foot–ground interaction and postural adaptation after downhill walking is compromised, albeit modestly. Nevertheless, even if differences appear small, automatized step control is likely affected and could react in non-ideal ways to perturbations.

A limitation of the current study can be seen in the movement uniformity of the intervention. Downhill walking on a treadmill indoors provides objective speed and incline, yielding comparable eccentric loads for participants as an important tool in objective experimental designs. However, the smooth surface, constant decline, replicated foot positioning, and unchanged friction are quite different from descents outdoors in the mountains. Another limitation was the kind of movements studied to yield the outcome variables, which represented simplified test situations to assure reliable measurements. Observing the control of postural demands in more dynamic situations might reveal further important findings. A shortcoming of the study design might be that no control group for uphill walking was included. However, mild-effort uphill walking has been shown to not affect joint position sense in a previous publication [26].

## 5. Conclusions

Our results support the hypothesis that simulated downhill walking leads to a decline in leg dexterity and to a reduction in postural ability in the anterior–posterior direction. The underlying mechanism, we believe, is that eccentric contractions disrupt the muscle proprioception pathways necessary for the control of short-latency, dynamic foot–ground interactions. Our results suggest that hiking safety after prolonged downhill walking in the mountains—even on easy paths—might be improved by training leg dexterity in unstable conditions, being aware of this naturally occurring sensorimotor impairment, and stepping more carefully during descents.

## Figures and Tables

**Figure 1 ijerph-20-05424-f001:**
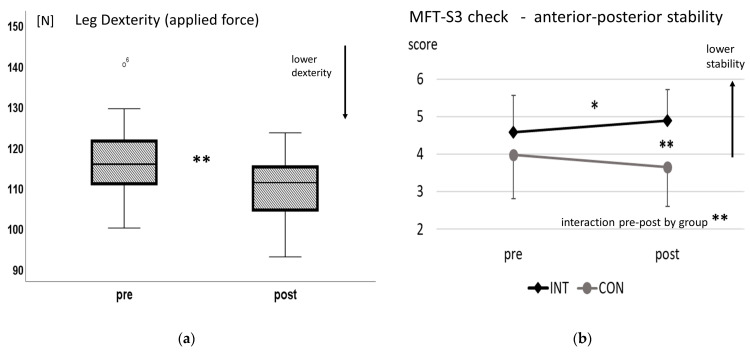
(**a**) Leg dexterity maximal force (boxplot) pre- and post-downhill walking (leg dexterity force in N), ** *p* < 0.01. (**b**) MFT-S3 balance test in anterior–posterior direction. Stability scores (mean and standard deviation) pre- and post-downhill walking for the intervention group (INT) and control group (CON), * *p* < 0.05, ** *p* < 0.01.

## Data Availability

The data presented in this study are available on request from the corresponding author.

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
