# Peer review of "Mountain Hiking: Prolonged Eccentric Muscle Contraction during Simulated Downhill Walking Perturbs Sensorimotor Control Loops Needed for Safe Dynamic Foot–Ground Interactions"

_ijerph, 2023, doi:10.3390/ijerph20075424_

Round 1

Reviewer 1 Report

An article by Werner et al. presents an alternative hypothesis of the cause of injury while walking downhill during tourism, which manifests itself in the form of reduced dexterity and imbalance of the legs and is usually attributed to fatigue.

In rather simple experiments on walking on a treadmill with a relatively slight incline, followed by an assessment of changes in agility and balance of the legs, the authors conclude that there is a violation of the control of the interaction of the legs not depending on fatigue. The degree of fatigue was controlled by heart rate and assessment of the well-being of the subjects on the Borg scale (RPE).

After 30 minutes of walking down, a significant decrease in leg dexterity was found, according to the authors. They also note a significant increase in the heart rate and RPE in the range of low effort. In my opinion, such increases can hardly be considered significant. Based on experimental data, they conclude that eccentric contractions disrupt muscle pathways proprioception needed for control of dynamic foot-ground interactions. Finally, they recommend performing training of the kind used in such experimental unstable conditions as a way to make mounting hiking safer, particularly downhill walking.

As mentioned by the authors, their experimental conditions are too ideal to provide real training. In reality, mounting hiking paths are much less smooth and comfortable than the treadmill. 

The suggested mechanism is rather a supplement but not an alternative to fatigue as a reason for hiking traumatism. Do not authors think over comparing the effects of downhill walking with that of uphill, where fatigue contribution is more pronounced? Such a comparison could give a piece of information in favor of one or another explanation or work out an integrated view of the injury mechanism.

The text contains multiple misprints and needs careful proofreading.

For instance: 

line 95: heart, not heat rate;

line 81: dropout

line 83: gender-balanced

etc

Author Response

An article by Werner et al. presents an alternative hypothesis of the cause of injury while walking downhill during tourism, which manifests itself in the form of reduced dexterity and imbalance of the legs and is usually attributed to fatigue.

In rather simple experiments on walking on a treadmill with a relatively slight incline, followed by an assessment of changes in agility and balance of the legs, the authors conclude that there is a violation of the control of the interaction of the legs not depending on fatigue. The degree of fatigue was controlled by heart rate and assessment of the well-being of the subjects on the Borg scale (RPE).

After 30 minutes of walking down, a significant decrease in leg dexterity was found, according to the authors. They also note a significant increase in the heart rate and RPE in the range of low effort. In my opinion, such increases can hardly be considered significant.

Thank you for this notion. Of course, in the normal language use we would not say “significant rise” describing a small (but evident) increase in heart rate. Consequently, we decided to omit “significant” but to add the statistical p-value in the text.   

Based on experimental data, they conclude that eccentric contractions disrupt muscle pathways proprioception needed for control of dynamic foot-ground interactions. Finally, they recommend performing training of the kind used in such experimental unstable conditions as a way to make mounting hiking safer, particularly downhill walking.

As mentioned by the authors, their experimental conditions are too ideal to provide real training. In reality, mounting hiking paths are much less smooth and comfortable than the treadmill.

We agree with you that walking on a treadmill is easy and comfortable, but decline and speed of walking can be equally designed for all participants and – this was our approach - might elicit the mentioned effect of eccentric muscle action on instability of foot-ground interaction control.  

The suggested mechanism is rather a supplement but not an alternative to fatigue as a reason for hiking traumatism.

Thank you for underlining that this mechanism might be a partial effect, but it could be an explanation of stumbling and slipping events in the descent where victims report only low levels of fatigue (Faulhaber et al. 2020). We think that we stated that in lines 32 – 36 where we call it “other potential contributors”. We follow your advice to change the wording in the abstract in line 13 from ”alternative hypothesis” to “supplementary hypothesis”.

Do not authors think over comparing the effects of downhill walking with that of uphill, where fatigue contribution is more pronounced? Such a comparison could give a piece of information in favor of one or another explanation or work out an integrated view of the injury mechanism.

Thank you for asking this question. Of course, we ran pilot studies regarding this topic and found that the effect on balance in the anterior-posterior direction turned out significant after short bouts of downhill walking and was less evident after short bouts of uphill walking. In addition, an effect was shown in the change of the joint position sense error after downhill walking compared to uphill walking (Bottoni et al. 2014). This implicated our assumption that the eccentric activity of muscles plays a role when fatigue is scarcely present (light RPE), and therefore designed further studies only regarding downhill walking. We make this point visible in the limitations.

The text contains multiple misprints and needs careful proofreading.

We are grateful to this comment. The revised document was now proofread by a native speaker.

For instance: 

line 95: heart, not heat rate;  done

line 81: dropout   done

line 83: gender-balanced   done

etc

Reviewer 2 Report

This manuscript mainly addresses the precautions one should take in order to avoid/lower the chances while hiking downhill, with a valid experimental design that concludes in the reasonable (and somewhat empirical knowledge) that when hiking downhill we should “mind our step” with a possible scientific explanation accounted to the forces that apply to the foot when stepping down to a hill, independently of possible fatigue. Authors claim that their findings would help minimize risks of injury when hiking, especially in the descent.

Strengths: I find the experimental design quite solid. The authors had previous pilot studies performed and power analysis which provided them with adequate information for the number of participants and the necessary measurements that were adequately performed to address their scientific question.

Weaknesses: On the other hand, their experiments were performed in gym-like environment that is quite different from actual situation seen in the outdoors eg a mountain with probable wind and rough landscape, muddy etc. (this is acknowledged by the authors themselves). Nevertheless, the final outcome of the study, which is well presented, is justified and in accordance with previous studies.

 Overall, it is an interesting paper, though it is not adding that much “new” knowledge to the field, but with a valid scientific design and the arguments made provide scientific data that can strengthen further an empirical knowledge for safety measurements when hiking.

I do not have recommendations for improvement, my recommendation was to publish as it is.

Interesting work, that can contribute novel knowledge to the field.

Author Response

We are pleased about your positive comments! Thank you for taking the time to review our manuscript!

Reviewer 3 Report

General comment: the manuscript is well writtent and the study have interesting potential repercussion. However a major concern is related to the design of the study, that under the present form, could not accurately test de influence of the mode of muscle contraction during decline walking (see specific comment). Some statements should also be amended to avoid misinterpretation and confusions.

Experimental design:

L 93: How could the authors ascertain their protocol would avoid the occurrence of fatigue? Please precise which fatigue is mentioned here, and provide reference to support the absence of fatigue.

The absence of a level or incline walking condition makes it impossible to test the influence that the eccentric mode of muscle contraction during decline walking exerted on sensorimotor control. Could the authors provide explanations about their choice not to include a control condition to decipher between the impact of neuromuscular fatigue or the influence of the eccentric mode of muscle contraction?

Discussion:

L 175: HR could not be ascribed to central fatigue, please remove.

L 176-177: I am surprise to read a statement about the absence of neuromuscular fatigue. What represent therefore the leg dexterity maximal score (in Newton)? Please amend to avoid misinterpretation.

L 182-183: due to the protocol and the measurement performed in the present study, authors could rather state that exercise induced a decrease in performance function that could be explained by alteration in neural efficiency (but with limitation given the absence of electrophysiological measurements).

Author Response

General comment: the manuscript is well written and the study have interesting potential repercussion. However a major concern is related to the design of the study, that under the present form, could not accurately test de influence of the mode of muscle contraction during decline walking (see specific comment). Some statements should also be amended to avoid misinterpretation and confusions.

Thank you for taking your time to review this paper and provide helpful comments!

Experimental design:

L 93: How could the authors ascertain their protocol would avoid the occurrence of fatigue? Please precise which fatigue is mentioned here, and provide reference to support the absence of fatigue.

We agree that a detailed definition of fatigue beyond an individual feeling of being fatigued is missing in our manuscript. Referring to low level perceived fatigue (see introduction and Faulhaber et al. 2020) we applied RPE scales, monitored heart rate and checked changes in maximum strength of knee extensors. We measured heart rates far below aerobic thresholds and light perceived effort (mean of 12.5) which can be translated to lactate levels lower than the 3mmol/l threshold (Scherr et al. 2013). In addition, we did not observe any loss in maximum strength capacity after the downhill walking bout, and therefore concluded that “no fatigue” was present. To avoid the use of an undefined “fatigue” we changed to the more precise notion “to assure low exercise intensity” referring to the paper of Scherr et al. (2013) in line 93.

The absence of a level or incline walking condition makes it impossible to test the influence that the eccentric mode of muscle contraction during decline walking exerted on sensorimotor control. Could the authors provide explanations about their choice not to include a control condition to decipher between the impact of neuromuscular fatigue or the influence of the eccentric mode of muscle contraction?

We appreciate this comment to clarify.
For the leg dexterity (LD) the condition of level walking (see line 201) and 30 min of rest did not affect LD test outcome (Nagamori et al. 2016). We now added this notion in line 201 to make the comparison to level walking more visible. You are right that the inclined walking was not performed, but this is due to results of our pilot studies and a study observing joint position sense.
In the pilot studies, we found that the effect on balance in the anterior-posterior direction was less evident after short bouts of uphill walking. In addition, an effect was already shown in the change of the joint position sense error after downhill walking compared to uphill walking (Bottoni et al. 2014). This implicated our assumption that the eccentric activity of muscles plays a role when fatigue is scarcely present (light RPE), and therefore designed further studies only regarding downhill walking and possible mechanisms. We added information in the limitation sector.

Discussion:

L 175: HR could not be ascribed to central fatigue, please remove. done

L 176-177: I am surprise to read a statement about the absence of neuromuscular fatigue. What represent therefore the leg dexterity maximal score (in Newton)? Please amend to avoid misinterpretation.

Thank you for this comment. To avoid misinterpretation we added information that unchanged maximal knee extension strength after the downhill walking bout indicates that leg dexterity measurement is not biased by any changes in maximum muscle contraction. The leg dexterity maximal score depends on the control of the instability of the spring where the force is applied – maximum forces achieved in the LD test are far lower than isometric maximum contractions.

L 182-183: due to the protocol and the measurement performed in the present study, authors could rather state that exercise induced a decrease in performance function that could be explained by alteration in neural efficiency (but with limitation given the absence of electrophysiological measurements).

Thank you for your hint to be careful with our interpretation when EMG measurements are missing. Therefore, we follow your suggestion to formulate the possibility of explanation by the outcome of our test results. We rephrased line 184, 185 (now 192-193).

Reviewer 4 Report

I want to thank the Editor and the Authors for the chance to review the present manuscript. In this research, the Authors aimed to measure leg dexterity, bipedal balance, and maximal leg extension strength before and after 30 minutes of simulated downhill walking.

The paper is overall well designed, and the presentation is clear. I compliment the Authors for the conception of such a brilliant study. However, minor modifications or explanations are needed to improve the manuscript further.

The detailed comments are listed below.

Line 34: when did the participants of the cited study actually rate their perceived fatigue? The timing of the rating may affect the reported value as diverse phycological mechanisms may intervene (e.g., recovery, etc.). Please specify.

Line 63: the proposed simulated downhill walking is assumed to be a “non-fatiguing” task. Why did the Authors assume this characteristic? A better definition in the introduction section of what fatigue actually represents, along with its effects and supporting references, may support this statement and the proposed characterization of this specific exercise.

Lines 63-65: is this statement speculation from the Authors, or do previously published studies support it? In both cases, it is worth clarifying or adding supporting references.

Line 69: what is meant by “when fatigue is not detectable”? In this context, how is it assumed that fatigue may be detected?

Line 70: please justify the rationale for including two experiments with separate cohorts.

Line 73; I am not sure RPE could be a straightforward method to assess “central” fatigue as it does not directly measure the physiologic mechanisms responsible for central fatigue. In this case, “perceived”, “reported”, or “subjective” may be more appropriate.

Lines 77-78: how were these “expected” effect sizes computed from preliminary studies?

Lines 99-122: a figure depicting the Leg Dexterity test procedures may be appropriate to make them more transparent, as the explanation sounds confusing at certain points of the paragraph. Moreover, it is not clear the LD measurement unit. Is it quantified in Newton? Please specify.

Line 128: how is the stability score computed in this test?

Line 180: please recall and explain in the discussion section the meaning and action of this “underlying neural mechanism”.

Line 186: how is an exercise defined to be “non-fatiguing”? Which metrics or effects are considered to make the exercise actually fatiguing?

Author Response

I want to thank the Editor and the Authors for the chance to review the present manuscript. In this research, the Authors aimed to measure leg dexterity, bipedal balance, and maximal leg extension strength before and after 30 minutes of simulated downhill walking.

The paper is overall well designed, and the presentation is clear. I compliment the Authors for the conception of such a brilliant study. However, minor modifications or explanations are needed to improve the manuscript further.

The detailed comments are listed below.

Thank you for taking your time to review our manuscript!

Line 34: when did the participants of the cited study actually rate their perceived fatigue? The timing of the rating may affect the reported value as diverse phycological mechanisms may intervene (e.g., recovery, etc.). Please specify.

We agree that this is an important information. We specified to “at the beginning” – this was after the first five minutes – and “immediately at the end” of the walking bout just before leaving the treadmill.

Line 63: the proposed simulated downhill walking is assumed to be a “non-fatiguing” task. Why did the Authors assume this characteristic? A better definition in the introduction section of what fatigue actually represents, along with its effects and supporting references, may support this statement and the proposed characterization of this specific exercise.

Thank you for indicating this lack of definition in our manuscript. Fatigue itself is defined as a decrement of observed variables – for example in eccentric exercises where the MVC measurement is controlled to achieve a decrement of 40 to 50% (fatiguing protocol) and then measured e.g., joint position sense. We controlled HR and RPE as well as maximum strength to characterize our exercise. As RPE has been shown to be highly correlated with HR and lactate levels (Scherr et al. 2013) we can state that in our exercise athletes did not reach HR near the aerobic threshold as well as RPE scores around 12.5 where lactate levels under 3mmol/l are expected. In combination with the lack of strength loss in knee extensors after the bout, we conclude that our exercise is a “non fatiguing” task for the participants. Following your comment, we avoided the term “fatigue” when specifying our exercise (line 66, 97, 181).

Lines 63-65: is this statement speculation from the Authors, or do previously published studies support it? In both cases, it is worth clarifying or adding supporting references.

This is an assumption of the authors and not being studied yet. Our hypotheses emerge on observations made in different studies (Nagamori et al. 2016, Bottoni et al. 2014 (added) and Faulhaber et al. 2020). We added these references there.

Line 69: what is meant by “when fatigue is not detectable”? In this context, how is it assumed that fatigue may be detected?

This notion should underpin the idea that not the common kind of “fatigue” – loss of strength, high effort exercise leading to exertion – is exclusively responsible for changes in motor performance (which has been shown already) but rather this described mechanism even if mountain hikers are not fatigued in the sense mentioned above. We rephrased this sentence.

Line 70: please justify the rationale for including two experiments with separate cohorts.

Thank you for this question. As we expected that after-effects will be disrupted by different activities we decided to separate the two tests, LD and MFT. In addition, we added the maximum strength measurement. In order to avoid adaptation to the exercise and to keep the participation within a justifiable time required for the experiments, we decided to conduct the study with two separate cohorts.

Line 73; I am not sure RPE could be a straightforward method to assess “central” fatigue as it does not directly measure the physiologic mechanisms responsible for central fatigue. In this case, “perceived”, “reported”, or “subjective” may be more appropriate.

Thank you for this comment – we replaced “central fatigue” with subjective fatigue

Lines 77-78: how were these “expected” effect sizes computed from preliminary studies?

We used Cohen’s d and partial eta squared out of preliminary studies.

The word “expected” should express that sample size can only be estimated relying on the assumption of constant effect sizes. Therefore, these calculated effect sizes were “expected” to occur in the new experiment.

Lines 99-122: a figure depicting the Leg Dexterity test procedures may be appropriate to make them more transparent, as the explanation sounds confusing at certain points of the paragraph. Moreover, it is not clear the LD measurement unit. Is it quantified in Newton? Please specify.

Thank you for this advice, we added Newton in Line 119 (now 121). Different figures and further information better explaining the test can be found in the references, especially please see Lyle et al. 2013 (http://dx.doi.org/10.1016/j.jbiomech.2012.11.058)

Line 128: how is the stability score computed in this test?

A tilt sensor is mounted on the platform and samples the tilt with a rate of 100 Hz. For the stability score tilt position and changes in tilt position over time are summed up and converted to the stability score. The less stable the platform position the higher the score.

Line 180: please recall and explain in the discussion section the meaning and action of this “underlying neural mechanism”.

Thank you for this feedback of missing clarity. The following sentence was rephrased to better see the connection. The neural mechanism is explained again later in the discussion (line 226 f)

Line 186: how is an exercise defined to be “non-fatiguing”? Which metrics or effects are considered to make the exercise actually fatiguing?

We would have thought to use the wording “non-fatiguing” for the expected perceived low load in downhill walking. No big changes in HR as well as in RPE as well as no change in maximum muscle strength in the most important muscle acting in our motor performance testing was observed. To avoid the use of not well-defined terms we omitted the use of “non-fatiguing”.

Round 2

Reviewer 1 Report

The text needs to be proofread once again: lost articles, commas, etc.

An example: L257 - legs, not lex, I assume.

Author Response

The text needs to be proofread once again: lost articles, commas, etc.

Thank you for the precise check of our text once again! We now concentrated on spell check, articles and commas.

An example: L257 - legs, not lex, I assume.  changed to “leg”, thank you for detecting this mistake.

Reviewer 3 Report

I thanks the authors for taking into consideration my comments. Authors clarified their manuscript as I requested.

Author Response

I thanks the authors for taking into consideration my comments. Authors clarified their manuscript as I requested.

Thank you for your helpful comments and your positive assessment.